# The Biological Properties of the SARS-CoV-2 Cameroon Variant Spike: An Intermediate between the Alpha and Delta Variants

**DOI:** 10.3390/pathogens11070814

**Published:** 2022-07-20

**Authors:** Stefano Pascarella, Martina Bianchi, Marta Giovanetti, Domenico Benvenuto, Alessandra Borsetti, Roberto Cauda, Antonio Cassone, Massimo Ciccozzi

**Affiliations:** 1Dipartimento di Scienze Biochimiche A. Rossi Fanelli, Università degli Studi di Roma La Sapienza, 00185 Roma, Italy; stefano.pascarella@uniroma1.it (S.P.); martina.bianchi@uniroma1.it (M.B.); 2Laboratory of Flavivirus, Oswaldo Cruz Institute, Oswaldo Cruz Foundation, Rio de Janeiro 21040-900, Brazil; giovanetti.marta@gmail.com; 3Department of Science and Technology for Humans and the Environment, University of Campus Bio-Medico di Roma, 00185 Rome, Italy; 4Faculty of Medicine, University of Campus Bio-Medico di Roma, 00185 Rome, Italy; domenicobenvenuto95@gmail.com; 5Istituto Superiore di Sanità, 00185 Rome, Italy; alessandra.borsetti@iss.it; 6Fondazione Policlinico Universitario Agostino Gemelli IRCCS, 00168 Roma, Italy; roberto.cauda@unicatt.it; 7Universita degli Studi di Siena—Sede di Arezzo, 52100 Arezzo, Italy; antonio.cassone2@gmail.com

**Keywords:** B.1.640.2, B.1.640.1, IHU, Alpha variant, Delta variant, NTD, net charge

## Abstract

An analysis of the structural effect of the mutations of the B.1.640.2 (IHU) Spike Receptor Binding Domain (RBD) and N-terminal Domain (NTD) is reported along with a comparison with the sister lineage B.1.640.1. and a selection of variants of concern. The effect of the mutations on the RBD–ACE2 interaction was also assessed. The structural analysis applied computational methods that are able to carry out in silico mutagenesis to calculate energy minimization and the folding energy variation consequent to residue mutations. Tools for electrostatic calculation were applied to quantify and display the protein surface electrostatic potential. Interactions at the RBD–ACE2 interface were scrutinized using computational tools that identify the interactions and predict the contribution of each interface residue to the stability of the complex. The comparison among the RBDs shows that the most evident differences between the variants is in the distribution of the surface electrostatic potential: that of B.1.640.1 is as that of the Alpha RBD, while B.1.640.2 appears to have an intermediate surface potential pattern with characteristics between those of the Alpha and Delta variants. Moreover, the B.1.640.2 Spike includes the mutation E484K that in other variants has been suggested to be involved in immune evasion. These properties may hint at the possibility that B.1.640.2 emerged with a potentially increased infectivity with respect to the sister B.1.640.1 variant, but significantly lower than that of the Delta and Omicron variants. However, the analysis of their NTD domains highlights deletions, destabilizing mutations and charge alterations that can limit the ability of the B.1.640.1 and B.1.640.2 variants to interact with cellular components, such as cell surface receptors.

## 1. Introduction

The COVID-19 pandemic continues to have a threatening impact on a global scale, in part due to newly emerging SARS-CoV-2 variants, such as Delta and Omicron [1,2,3,4,5,6]. The continuous discovery of new variants requires tracking the diffusion of the newly mutated viruses in response to the concerns generated in the general population. The evaluation of the existing vaccines and the development of new ones are among the priorities as well as the monitoring of the vaccines’ effectiveness against the new variants. New COVID-19 variants have repeatedly emerged over the past year. However, only few of them have eventually been found more dangerous, mostly because of their higher transmissibility [2,3,7]. By the end of November 2021, a new variant was discovered in southern France, quickly resulting in a small outbreak [8]. Recently, the released data suggested that the labelled B.1.640.2 variant, then renamed IHU by the WHO (World Health Organization), seems to come from Cameroon. It is phylogenetically related to the ancestor B.1.640 lineage, which was later renamed B.1.640.1. Only few data are now available on this variant. Whether it is of concern could be determined as experts gain further understanding of how it spreads and causes illness. Although animal models and in vitro studies can provide important information, clinical and molecular data are needed to determine whether the existing vaccines and drugs are losing efficacy against the emerging variants. In this study, we shed light on the mutational pattern of this emerging variant with the aim of determining a possible biological tradeoff between viral transmission and immune escape as well as to better understand the vaccines’ effectiveness and virus contagiousness.

## 2. Results

### 2.1. Structural Analysis of the Spike N-Terminal Domain (NTD)

The Spike NTD possesses a lectin-like fold [9] and it is deemed to be involved in the interaction with surface cell components, such as AXL UFO receptor [10,11] and/or glycosphingolipids [12]. Moreover, it cooperates with RBD–ACE interactions during virus entry into the cell. NTD is also relevant as a target of the host immune response [13]. 

Mutations occurring at the NTD, with respect to the reference Wuhan virus, were compared to the characteristic mutations of a selection of the most widespread variants (Appendix A) of concern (VOC): Alpha, Delta and Omicron [2,3,4,6,14]. Mutations in IHU NTD are unique compared to those in the other considered variants (Appendix A). The two sister lineages differ from each other for the I210T and D215H mutations, which are unique to B.1.640.1 and B.1.640.2, respectively (Table 1 and Appendix A). Most notably, a 9-residue long deletion has been reported to occur in IHU B.1.640.2 and B.1.640.1 between sequence positions 136–144 (Figure 1). The mutations IHU and B.1.640.1 remove several stabilizing interactions (reported in Table 1). In particular, the wild-type salt bridge E96-R190 and the disulfide bridge C15-C136 are lost in both variants, while the salt bridge D215-R214 is removed only in IHU NTD (Figure 1). The impact of these mutations on the stability of the NTD has been assessed by the application of FoldX [15]. FoldX was applied to calculate the ΔΔG corresponding to each mutation using the PDB coordinate set 7B62 as the reference structure. The deletions of C136 within the deleted segment 136–144 were simulated as the substitution of Cys to Ala. Ala was chosen as a replacing residue in analogy with the alanine scanning procedure. Indeed, replacement with alanine removes all atoms past the β-carbon. Thus, the role of sidechain functional groups can be inferred. Moreover, alanine lacks unusual dihedral angle preferences at variance with glycine, for example, which could introduce conformational flexibility and alter the analysis [16]. Deletions cannot be tested by the PositionScan function of FoldX and for this reason were not considered in this paper. All the mutations are predicted to be destabilizing (Table 2) at variance with those predicted for the variants Delta and Omicron (Appendix A). It should be considered that, in addition to the destabilizing effect of the point mutations, the deletion of the region 136–144 removes a β-strand belonging to the β-hairpin, deemed to contribute to the interaction with the receptor AXL [10,11] (Figure 1).

Overall, the net charge that indirectly reflects the property of the surface electrostatic potential [2,17] was calculated by the program PROPKA3 [18]. Mutations in the Alpha and Delta variants do not significantly affect the net charge of the NTD that remains at a value close to 1.0 as in the Wuhan domain. A notable exception is Omicron, which has a negative charge due to the 214EPE insertion. On the contrary, the IHU mutations E96Q and D215H remove two negative charges that result in a more positive net charge equal to 3.7 (Appendix A). This effect is less pronounced in B.1.640.1. This charge variation can affect and alter interactions with negatively charged cellular components [19]. For example, it may be speculated that a more positively charged NTD can exert a stronger attraction for the negatively charged syalosides bound to the cellular membrane components and for AXL receptor that displays a negative surface electrostatic potential [10].

### 2.2. Structural Analysis of the Spike Receptor Binding Domain (RBD)

A comparison between the spike RBD of the two sister lineages B.1.640.1 and the IHU B.1.640.2 shows that the two spikes differ for the mutations F490R in the B.1.640.1 lineage and E484K and F490S in B.1.640.2 (Table 1 and Appendix A). The comparison with the selected VOC shows a gradient of increasing number of mutations from the early Alpha variant up to the IHU, except for Omicron RBD. Indeed, Alpha RBD has only one mutation, while IHU has six (Appendix A). 

The impact of these mutations on the stability of the NTD was assessed by the application of FoldX [18]. The mutations in B.1.640.1 and IHU seem to be constantly destabilizing, although with different weight (Appendix A). This is not the case of the other RBD variants, except for the Alpha RBD that has only one mutation. 

The RBD net charge at pH = 7.0 (the overall charge resulting from the combination of the charge of the single ionizable residue side chains) calculated by PROPKA3 is about 2.6 for Alpha and 2.2 for B.1.640.1, which is as that of the Wuhan spike (2.1). The variant IHU has, instead, a higher net charge of about 3.2. It should be noted that the predicted net charges of Delta (lineage B.1.617) and Omicron (lineage B.1.1.529) RBDs are higher, namely 4.1 and 5.2, respectively [2]. The charge similarity is also reflected by the similarity of the pattern of the electrostatic potential on the surface of Alpha and B.1.640.1 RBDs, while B.1.640.2 displays a pattern that is in between that of the two variants.

The electrostatic pattern between the Alpha and Delta variants is shown in Figure 2. The form and intensity of the surface electrostatic potential is interesting as it is considered one of the factors that can determine the stability and specificity of interaction RBD–ACE2 [3,20] and may contribute to the prediction of variant infectivity [20]. Other factors that contribute in general to the spike–receptor interaction are those observed at protein interfaces, namely van der Waals contacts, hydrogen bonds and salt bridges [21].

The ACE2 interaction pattern of the B.1.640.1 and IHU variants was compared in detail with that of the Alpha variant and resulted to be similar. Indeed, RING2.0 identifies similar interactions between Alpha, the two B.1.640 RBDs and ACE2, except for the site 449, where a Tyr is replaced by an Asn. In fact, in this case, Asn seems not to contribute any significant interaction (Figure 3 and Appendix A) to the interface stabilization.

F490 is replaced by an Arg that may form a weak salt bridge with ACE2 E35 in B.1.640.1, while it is replaced by Ser in B.1.640.2. However, B.1.640.2 displays the substitution E484K, absent in B.1.640.1, which has also been observed in the B.1.351 (Beta), P1 (Gamma) and B.1.621 (Mu) variants. K484 may form a weak salt bridge with ACE E75 and it is deemed to be involved in immune evasion [22]. 

Alanine scanning obtained through DrugScorePPI [23] confirms that the Alpha “hot-spots” are conserved in the B.1.640.1 and 2 interfaces. In particular, N487, Y489, Q493, Y501 and Y505 are the residues that offer the highest contribution to the interaction stabilization. Within these residues, Y501 and Y505 appear dominant. In this case, N449 of B.1.640.1 is also predicted not to be part of the interface with ACE2. Likewise, K484 is predicted not to be an interface hot-spot in the IHU RBD–ACE2 complex (Appendix A).

## 3. Discussion

The B.1.640.1 and IHU variants have had a limited diffusion. Structural comparison with more “successful” variants may indicate a few of the structural factors that may promote a higher infectivity. Based on the comparative analysis of the critical spike mutations, the B.1.640.1 appears to share biological similarity with the Alpha variant regarding infectivity and/or immune escape, while B.1.640.2 IHU shows partly different properties. 

The spike RBD net charge at pH = 7.0 is about 2.2 for Alpha and 2.6 for B.1.640.1, displaying a pattern as those of the Alpha variant [24]. Instead, B.1.640.2 has a predicted net charge at pH = 7.0 equal to about 3.2. The net charges of the Delta and Omicron variants RBDs are higher (4.1 and 5.2, respectively). Moreover, the charge surface distribution of IHU RBD is intermediate between that observable in Alpha and Delta RBDs. The pattern of interaction with ACE2 is also similar for Alpha and B.1.640.1 and 2, suggesting no great contagiousness as the Omicron variant. 

However, the analysis of the theoretical impact of the mutations into the RBD stability suggests that the mutation characteristics of IHU and B.1.640.1 are all destabilizing. Moreover, the structural analysis of NTD shows interesting patterns. The deletion in the region 136–144 removes a β-strand deemed to be involved in the interaction with the AXL receptor. Moreover, the characteristic mutations are significantly destabilizing. The overall NTD net charge is also markedly different from that of the other variants, being almost twice as much the charge of the Alpha and Delta NTDs. Omicron is an exception with its negative charge. In conclusion, although IHU and B.1.6401.1 present an RBD apparently as that of the Alpha and Delta variants, our analysis suggests that it is associated with an NTD that does not appear optimized in terms of the stability and recognition of cellular molecular components. This strongly supports the notion that the properties of RBD depend also on the characteristics of the cognate NTD that is an active and essential component of the spike and a target of immunological response [25]. 

Nevertheless, our study also showed that B.1.640.2 has probably lower infectivity than Delta but possibly higher than Alpha, implying a potential biological trade-off between viral transmission and immune escape. Indeed, the mutation E484K, absent in Alpha and B.1.640.1, is considered critical for immune escape [26]. 

Further, it is important to consider once again the impossibility to predict the likely emergence of potentially deadly new variants that might have a negative impact in the progression of the pandemic. 

Although these findings are theoretical and speculative, they are valuable for the rational design of experiments aimed at unveiling the structure–function relationships in different variants and possibly predict their contagiousness ab initio. Moreover, the reported observations suggest considering the analysis of the NTD along with RBD, which is an active and important component of the SARS-CoV-2 mechanism of infection.

## 4. Materials and Methods

Mutations of the spike variants were obtained from the website https://cov.lanl.gov (accessed on 1 July 2022), while lineage definition and epidemiological statistics were obtained from the websites https://cov-lineages.org (accessed on 1 July 2022) [27] and https://www.outbreak.info (accessed on 1 July 2022) [28]. These resources identify variants and variant mutations by the analysis of the GISAID database content [29]. All the websites were accessed in March 2022. The structure of the variant spike RBD (receptor-binding domain) was built by in silico mutagenesis using as a template the structure of the complex ACE2–spike corresponding to the PDB code 7SY1. This entry contains the Cryo-EM structure of the D614G, N501Y mutant spike ectodomain in complex with ACE2 solved at the resolution of 2.83 Å. The RBD–ACE2 complex was cut out from the entire Spike in the 7SY1 entry and was subsequently mutagenized. The variant NTD was modelled using the PDB structure 7B62 corresponding to the crystal structure of the NTD in complex with biliverdin. In silico mutagenesis was carried out with the ad hoc tools available within UCSF-Chimera [30] or PyMOL [31] graphics programs. Energy minimization protocol embedded in the molecular graphics program Swiss-PdbViewer [32] was applied to the model RBD to remove residue steric overlaps at the interface. The protocol used the GROMOS96 43B1 force field, cutoff 10 Å and 100 steps of steepest descent minimization followed by 1000 steps of conjugate gradients in vacuo. The minimization was stopped if the energy difference between two consecutive steps was lower than 0.05 kJ/mol. Only residues at the interface were minimized. This forcefield does not include the parameters to describe glycans, which were therefore ignored during minimization. However, the glycans present in the RBD are apparently far from the complex interface and have not the potential to interfere with it. 

Foldx5 [18] suite was applied to check and repair side chain with anomalous stereochemistry or energy using the RepairPDB function and to calculate the fold energy change upon point mutation using the PositionScan function [18]. The folding energy change (∆∆G) is expressed as
∆∆G = ∆G_mut_ − ∆G_wt_(1)
were ∆Gmut and ∆Gwt are the folding energy of the mutant and wild type protein, respectively. In this way, ∆∆G positive values indicate destabilizing mutations, while negative values the opposite.

Electrostatic potentials were calculated with the APBS tool [33] program and mapped onto the protein surface with PyMOL. The domain net charge, namely the overall charge resulting from the combination of the charge of the single ionizable residue side chains, was calculated by the program PROPKA3 [18]. This program predicts the pKa of ionizable residues within the local structural environment. This includes solvent exposure, hydrogen bonds or interactions with other charged groups. The calculated pKa values are combined to predict the overall net charge of the protein domain at a given pH. In this work, the value of the pH was set to 7.0 as a standard state for the comparison of the net charge of the different variant domains. This value does not necessarily reflect the physiological environment. 

The non-covalent interactions taking place at the interface of the predicted complexes were identified using the RING 2.0 web server [34] at the URL https://ring.biocomputingup.it/ (accessed on 1 July 2022). The server implements a fast tool to detect intra- and inter-chain interactions, including solvent and ions. The computational alanine scanning of the interface residues of the spike complexes was carried out through the webserver DrugScorePPI [23] available at the URL https://cpclab.uni-duesseldorf.de/dsppi/main.php (accessed on 1 July 2022). The server provides a fast and accurate tool to predict the binding free energy changes upon alanine mutations at protein–protein interfaces using a knowledge-based scoring function.

## 5. Conclusions

The structural analysis suggested that IHU spike shares similarities with the Alpha variant while displaying differences with Omicron. This observation prompts the definition of the IHU variant as a variant of interest (VOI) and not a variant of concern (VOC). According to the definition by the WHO, a VOI is a variant with genetic changes that are predicted or known to affect virus characteristics (transmissibility, disease severity, immune escape, diagnostic or therapeutic escape) and identified as causing a significant transmission or multiple COVID-19 clusters, which may suggest a potential risk for public health. A VOC is a VOI that displays increased transmissibility and/or worsening of COVID-19 epidemiology and/or increase in virulence and/or decrease in the effectiveness of the available diagnostics, vaccines and therapeutics. 

Nonetheless, it is important to consider once again the impossibility of predicting the emergence of a new variant introduction from abroad and then the difficulty of controlling its subsequent spread in a country. It is necessary to assure a rigorous genomic surveillance worldwide as an implementation of the genomic track system to prevent an eventual passage of this variant and the eventual spread in other regions. It is too early to speculate in this way, but we must be alert for this variant and others of concern that can be introduced in a country. Of course, vaccination and government advice must be considered by people to protect themselves and others.

## Figures and Tables

**Figure 1 pathogens-11-00814-f001:**
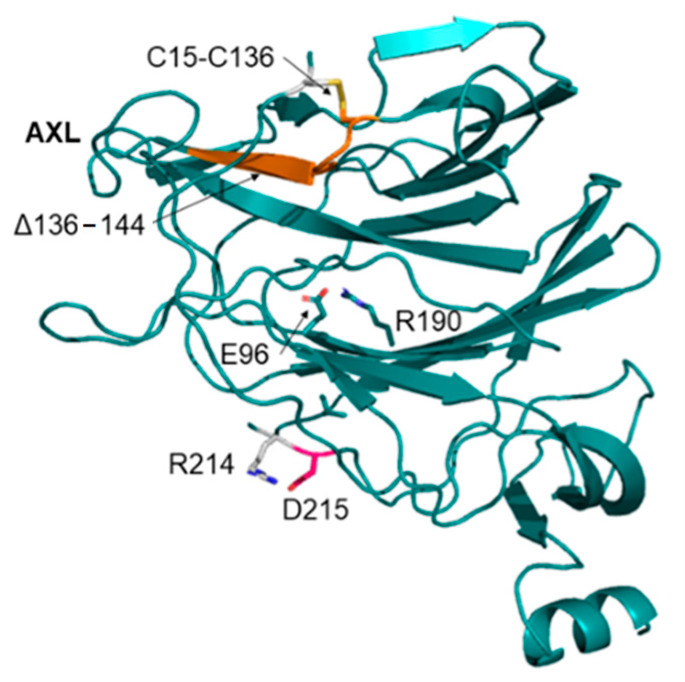
Cartoon model of the SARS-CoV-2 NTD reported in the PDB coordinate set 7B62. The orange portion indicates the deletion encompassed by the sequence positions 136–144. The label “AXL” marks the region predicted to interact with the AXL receptor. Disulfide bridge lost upon deletion is labelled and displayed as stick model. Side chains corresponding to the point mutations are depicted as stick models. Red sticks denote the D215 mutated site specific to the IHU variant, while the Arg partner in the salt bridge is colored white.

**Figure 2 pathogens-11-00814-f002:**
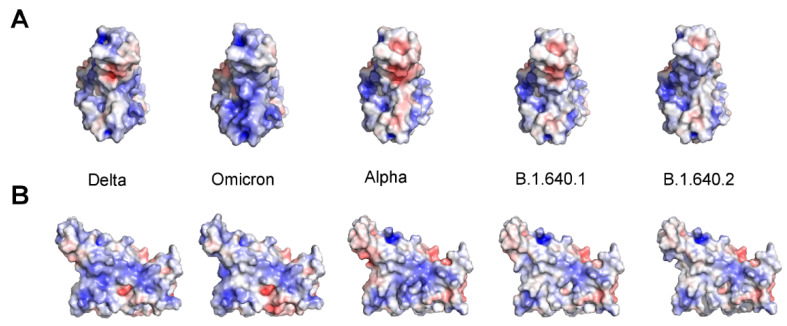
Comparison of the electrostatic potential surface of the Delta, Omicron and Alpha VOC and the B.1.640.1 and B.1.640.2 variants of the spike RBD. The red and blue colors indicate negative and positive potential, respectively. The color scale ranges from −5.0 to +5.0 kT/e. The RBD is oriented with the ACE2 interface in the front (**A**) or rotated 90° to the left along the y-axis (**B**).

**Figure 3 pathogens-11-00814-f003:**
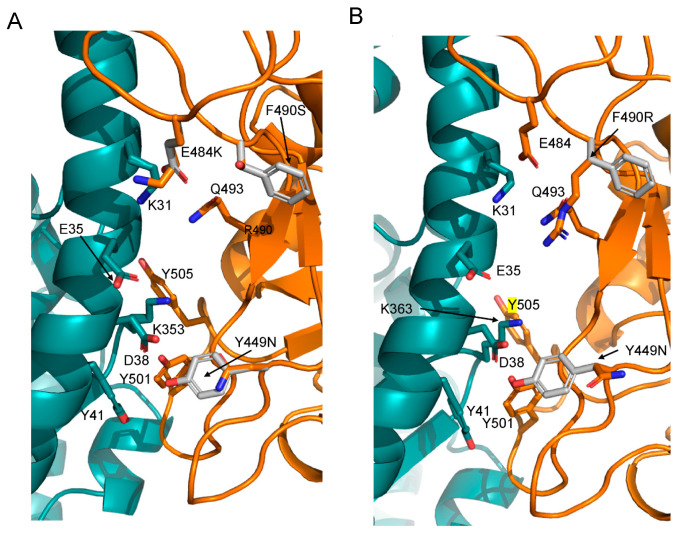
Predicted interactions at the interface between B.1.640.2 (**A**) and B.1.640.1 (**B**) RBDs and ACE2. ACE2 and spike RBD are displayed as deep teal and orange, respectively. The relevant side chains are represented as labelled sticks. The grey side chain represents the residue of the Alpha variant in the corresponding position. Their labels indicate the original and the replacing side chains.

**Table 1 pathogens-11-00814-t001:** Mutational pattern and structural context. B.1.640.2 mutations occurring in the Spike protein with respect to the reference Wuhan spike protein. Structural context and interaction changes (added or removed) compared to those found in the reference 7SY1 spike.

Mutations ^a^	Structural Context ^b^	Changed Interactions
P9L	N-terminal. Structurally unresolved	
E96Q	NTD; loop connecting two β-strands	Removes the salt bridge with R190
Δ136–144	NTD: strand of a β-hairpin	Removes a β-strand and the disulfide bridge C15-C136 Potentially interacting with AXL
R190S	NTD; within the β-strand encompassed by positions 188–197	Removes the salt bridge with E96
D215H	NTD: in an exposed loop	Removes salt bridge with R214
R346S	RBD: loop connecting two α-helices	
N394S	RBD: loop at the interface with the NTD of the other chain	
**Y449N**	RBD: loop connecting a short α-helix and a β-strand	Removes H-bond with ACE2 D38
**E484K**	RBD: within a loop	Possible weak salt bridge with ACE2 E75
**F490S**	RBD: within a loop near to K484	
**N501Y**	RBD	Interaction with ACE2 Y41 and K353
D614G	S1	
P681H	S1: Exposed loop not visible in the reference structure	
T859N	S2: within a β-strand at the interface with the other subunit	Forms a H-bond with N317 of the other subunit
D1139H	S2: N-terminus of an α-helix near to HR2	

^a^ ACE2 interface residues are boldfaced. ^b^ S1 and S2 refer to the receptor-binding fragment 1 and the fusion fragment 2 of the spike monomer, respectively.

**Table 2 pathogens-11-00814-t002:** Variation of the folding energy upon the mutation calculated by FoldX. Mutations in B.640.1 and IHU are reported.

Mutation	ΔΔG (kcal/mol)
E96Q	1.3
C136A	4.3
R190S	4.0
I210T	2.9
D215H	2.0

## Data Availability

Not applicable.

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
