# Peer review of "The Biological Properties of the SARS-CoV-2 Cameroon Variant Spike: An Intermediate between the Alpha and Delta Variants"

_pathogens, 2022, doi:10.3390/pathogens11070814_

Round 1

Reviewer 1 Report

In this paper, Stefano Pascarella et al. inspected the distribution of the surface charge of Delta, Omicron, Alpha, B.1.640.1, and B.1.640.2 variants of the Spike Receptor binding domain. The main research idea is interesting, nonetheless, obtained results seem to be too preliminary at this point, and the conclusion is not fully supported by the data. 

Major points:

- Introduction must be improved, and research aims need to be clearly stated. Overall, the paper should be better placed in the research context.

- Citing only 21 sources seems a little bit small to me (and 3 of them are auto citations), reference section should be improved to better cover current research in the field

- Authors state that "The form and intensity of the surface electrostatic potential is interesting as it is considered one of the factors that can determine the stability and specificity of interaction RBD-ACE2" ... this is true, but what about the other factors? The authors' conclusions are too bold claims and not fully supported by analyses they carried out

- I strongly suggest adding some complementary bioinformatic analyses to better support the authors' claims

- Discussion section needs to be improved and they should discuss obtained results more extensively

Minor points:

Line 16 ... RBD (and all other abbreviations) needs to be explained

Line 37 ... should be "transmissibility"

Line 43 ... in vitro, in silico, etc., should be always in Italics

Line 44 ... what is "your country"? Italy, Brasil, or something else? Such expressions should be replaced, Pathogens is an international journal

Line 59 ... by Propka (?)

Line 80 ... I see nothing blue here

Line 103 ... the B.1.640.1 is likely to share biological similarity ... too bold claim not supported by the data

Line 111 ... Spyke

Methods section ... always add tool web addresses into the brackets, and the date they were accessed

Line 150 ... In Italy

Author Response

Reviewer 1

General Comment: In this paper, Stefano Pascarella et al. inspected the distribution of the surface charge of Delta, Omicron, Alpha, B.1.640.1, and B.1.640.2 variants of the Spike Receptor binding domain. The main research idea is interesting, nonetheless, obtained results seem to be too preliminary at this point, and the conclusion is not fully supported by the data. 

Reply: The manuscript reports a theoretical analysis of the characteristics of the RBD of the IHU Spike and compare them to other main variants. Conclusions should be tested by experiments. We agree with the referee. Therefore, we have extended our analysis to the NTD domains and applied FoldX in addition to the other tools to test the impact of mutations on the stability of RBD and NTD. We have discovered that IHU NTD is destabilized more than the other variants and that it contains a deletion that may weaken interaction with cell surface receptors such as AXL. These further analyses should better support our conclusions. A new Figure 1 has been included, Table 1 modified and Table 2 added. Supplementary Tables have been updated and S1 and S3 added. In agreement with the review we like to remark that our analysis are founded on mathematical and theoretical model and sometimes speculate on the results giving important information for successfully lab analysis on cell and animal to confirm our results and hypothesis.

Major points:

Comment 1: Introduction must be improved, and research aims need to be clearly stated. Overall, the paper should be better placed in the research context.

Reply: Introduction has been revised in the light of the new analyses that have been carried out

Comment 2: Citing only 21 sources seems a little bit small to me (and 3 of them are auto citations), reference section should be improved to better cover current research in the field

Reply: We have expanded the number of citations that now are 37

Comment 3: Authors state that "The form and intensity of the surface electrostatic potential is interesting as it is considered one of the factors that can determine the stability and specificity of interaction RBD-ACE2" ... this is true, but what about the other factors? The authors' conclusions are too bold claims and not fully supported by analyses they carried out

Reply: We have added a more explicit sentence with a reference

Comment 4:  I strongly suggest adding some complementary bioinformatic analyses to better support the authors' claims

Reply: Completely agree with the referee. We have expanded the analysis to the NTD domain and applied FoldX for structure regularization and destabilization analysis

Comment 5:  Discussion section needs to be improved and they should discuss obtained results more extensively

Reply: Yes, we agree. The discussion has been changed according to the new analysis and results.

Minor points:

Comment 6: Line 16 ... RBD (and all other abbreviations) needs to be explained

Reply: Done. Definitions are in the Abstract and in the Introduction

Comment 7: Line 37 ... should be "transmissibility"

Reply: Done

Comment 8: Line 43 ... in vitro, in silico, etc., should be always in Italics

Reply: Done

Comment 9: Line 44 ... what is "your country"? Italy, Brasil, or something else? Such expressions should be replaced, Pathogens is an international journal

Reply: Agree. We have modified the sentence as “In many countries worldwide, including Italy, efforts to track viral mutations and variants are ongoing.

Comment 10: Line 59 ... by Propka (?)

Reply: Done

Comment 11: Line 80 ... I see nothing blue here

Reply: Thanks for pointing this out. The correct color definition is “deep teal”. We have modified the caption accordingly

Comment 12: Line 103 ... the B.1.640.1 is likely to share biological similarity ... too bold claim not supported by the data

Reply: The discussion has been revised considering the new results and analyses.

Comment 13: Line 111 ... Spyke

Reply: Thanks for pointing this out. The sentence (and the uncorrectly spellt Spyke) has been removed.

Comment 14: Methods section ... always add tool web addresses into the brackets, and the date they were accessed.

Reply: Done

Comment 15: Line 150 ... In Italy

Reply: We apologize. We are not sure we correctly understood this point. In any case, we have modified the sentence. We hope it is now OK.

Reviewer 2 Report

The article has a title that initially seems attractive. When one begins to read it, one appreciates that it is somewhat superficial, not very extensive in the topics it exposes. 

Most of the bibliography it provides is outdated and there are even some quotations that are poorly referenced.

I think that the work should be redone trying to offer a greater depth in the topics that it exposes and likewise improve the global presentation.

Author Response

Reviewer 2

General comment: The article has a title that initially seems attractive. When one begins to read it, one appreciates that it is somewhat superficial, not very extensive in the topics it exposes. 

Reply: We have deepened our analysis by focusing the attention also on the NTD portion of the Spike. Moreover, we have tested the impact of the mutations using FoldX, one of the state-of-the-art programs available. A new Figure 1 has been included, Table 1 modified and Table 2 added. Supplementary Tables have been updated and S1 and S3 added. We believe that now this article can give important information between the interaction RBD /NTD very important to understand the infectiveness of this variant.

Comment 1: Most of the bibliography it provides is outdated and there are even some quotations that are poorly referenced.

Reply: The bibliography has bee expanded up to 37 references.

Comment 2: I think that the work should be redone trying to offer a greater depth in the topics that it exposes and likewise improve the global presentation.

Reply: The manuscript has been extensively revised and now reports further hypothesis based on the new results. We hope that this now makes out manuscript more interesting. The aim of our theoretical analysis is to suggest rational hypotheses to the researchers that may help them designing experiments and tests.

Round 2

Reviewer 1 Report

The authors have improved the manuscript significantly. Nonetheless, there are still some issues that must be addressed before the publication:

1.) The title of the article is "The biological properties of the SARS CoV-2 Cameroon variant: an intermediate between Alpha and Delta variants" ... nonetheless only Spike protein was inspected in this study ... the title should be adequately edited to better express the content of the paper

2.) Abstract should be improved as well to be more reader-friendly. The abstract should have a similar structure as the whole article (IMRaD)

Line 33 ... lead to track the emergence

Line 37 ... only fe

Line 45 ... in vitro should be in Italics

Line 47 ... shield light should be shed light (?)

Results ... subsection headings should be formulated more comprehensively

Line 71 ... why Ala substitution and not e.g. Gly? Should be explained

Lines 92-93 ... "This charge change can affect and alter interactions with negatively charged cellular components [22]." ... this should be discussed more

Line 105 ... net charge parameter should be briefly explained to the readers

Line 105 ... why pH 7.0? It should be explained, as extracellular pH values differ significantly in various tissues

Lines 109 - 113 are very difficult to follow and should be carefully reformulated to be clear

Table 1 ... what do S1 and S2 abbreviations mean?

Line 232 ... "and possibly predict ab-initio of their contagiousness" ... ab initio should be in Italics, and overall it should be reformulated "and possibly predict their contagiousness ab initio"

Line 239 ... data base should be database

Line 241 ... in-silico should be in silico

Line 252 ... in vacuo should be in vacuo

The conclusion section is written very poorly ... "consider" word is repeated many times here ... in addition what's the difference between VOI and VOC? Should be briefly explained. Finally, why you are mentioning Italy again? Pathogens is an international scientific journal, and your study is not related to SARS-CoV-2 specifically in Italy, please reformulate

Author Response

Dear Editor,

We are grateful for the thoughtful comments of the referees. We have responded to all of them, and a point-to-point list is reported below. We hope that now the manuscript is acceptable for publication.

Many thanks,

Referee # 1

The authors have improved the manuscript significantly. Nonetheless, there are still some issues that must be addressed before the publication:

1.) The title of the article is "The biological properties of the SARS CoV-2 Cameroon variant: an intermediate between Alpha and Delta variants" ... nonetheless only Spike protein was inspected in this study ... the title should be adequately edited to better express the content of the paper

We have made the title more specific: now it explicitly mentions the Spike

2.) Abstract should be improved as well to be more reader-friendly. The abstract should have a similar structure as the whole article (IMRaD)

We have expanded the abstract trying to reproduce as much as possible the structure of the manuscript within the space limits. English also has been revised.

Line 33 ... lead to track the emergence

The sentence has been modified and we hope that now it is correct

Line 37 ... only fe

Thanks for pointing this. Corrected.

Line 45 ... in vitro should be in Italics

Done

Line 47 ... shield light should be shed light (?)

Thanks for pointing this out. Corrected

Results ... subsection headings should be formulated more comprehensively

Subheading have been made more explicit.

Line 71 ... why Ala substitution and not e.g. Gly? Should be explained

The reason for Ala substitution is well known to people involved in protein structural analysis. However, to clarify the motivation to a broader readership we have added a concise explanation and a reference for further details.

Lines 92-93 ... "This charge change can affect and alter interactions with negatively charged cellular components [22]." ... this should be discussed more

We have expanded the sentence.

Line 105 ... net charge parameter should be briefly explained to the readers

We have added an explanation in the Materials and methods section and at the text line indicated by the referee.

Line 105 ... why pH 7.0? It should be explained, as extracellular pH values differ significantly in various tissues

We have stated in the manuscript Materials and method section that “The value 7.00 of the pH has been set as a standard state for comparison of the net charge of different variant domains. The value does not necessarily reflect the physiological environment”.

Lines 109 - 113 are very difficult to follow and should be carefully reformulated to be clear

We have improved the sentence as far as possible.

Table 1 ... what do S1 and S2 abbreviations mean?

A definition has been added as a note to Table 1

Line 232 ... "and possibly predict ab-initio of their contagiousness" ... ab initio should be in Italics, and overall it should be reformulated "and possibly predict their contagiousness ab initio"

Thanks. Done

Line 239 ... data base should be database

Done

Line 241 ... in-silico should be in silico

Done

Line 252 ... in vacuo should be in vacuo

The conclusion section is written very poorly ... "consider" word is repeated many times here ... in addition what's the difference between VOI and VOC? Should be briefly explained. Finally, why you are mentioning Italy again? Pathogens is an international scientific journal, and your study is not related to SARS-CoV-2 specifically in Italy, please reformulate

We have tried to improve the conclusion: the style has been modified following the referee’s indications; VOI and VOC have been briefly defined and the references to Italy have been removed. We hope that now the conclusions are acceptable.

Reviewer 2 Report

After the revision suggested by the reviewers, the article has improved and can be considered for publication in the journal with the modifications that have been made. 

Author Response

Referee #2

After the revision suggested by the reviewers, the article has improved and can be considered for publication in the journal with the modifications that have been made. 

We are grateful for the appreciation of our work